# Factors driving underweight, wasting, and stunting among urban school aged children: Evidence from Merawi town, Northwest Ethiopia

Tilahun Tewabe[1], Md. Moustafa Kamal[2]*, Khorshed Alam[3], Ali Quazi[4], Majharul Talukder[4], Syeda Z. Hossain[2]

1 College of Medicine and Health Science, Bahir Dar University, Bahir Dar, Ethiopia, 2 Faculty of Medicine and Health, The University of Sydney, Sydney, Australia, 3 School of Business, Faculty of Business, Education, Law & Arts, and Centre for Health Research, University of Southern Queensland, Toowoomba, Queensland, Australia, 4 Canberra Business School, Faculty of Business, Government and Law, University of Canberra, Canberra, Australia

* mkamalanu@gmail.com

**Data Availability Statement:** The data of this study cannot be shared publicly due to the presence of sensitive (confidential) participants' information.

## Abstract

Prior research identified malnutrition as one of the most common causes of morbidity and mortality among children globally. Furthermore, research revealed that over two thirds of deaths associated with inappropriate feeding practices occurred during the early years of life. Improper feeding practices impact a child's health in many different ways. However, research on the possible factors driving underweight, wasting, and stunting among school aged children in developing countries is limited, hence warrant further attention. Against this backdrop, this research strives to identify and assess the determinants of underweight, wasting and stunting among school aged children of a developing country-Ethiopia. A community based cross-sectional study was conducted from April 1, 2018 to June 15, 2018 in Merawi town, Ethiopia. An interviewer-administered questionnaire was used to collect data from a sample of 422 children. Binary logistic regression technique was performed to examine the effect of each selected variable on the outcome measure. The prevalence of being underweight, wasting and stunting was found to be 5.7%, 9.8%, 10.4%, respectively. The age of the child [adjusted odds ratio (AOR) = 12.930 (2.350, 71.157)] and the number of children [AOR = 8.155 (1.312, 50.677)] were emerged as the key determinants for underweight, and the gender of the child was significantly associated with wasting [AOR = 0.455 (0.224, 0.927)]. Finally, the age of the child [AOR = 12.369 (2.522, 60.656)] was found to predict the risk of stunting. This study revealed the age, number of children and gender of the child to have a significant association with malnutrition. The findings of this research suggest that in improving the feeding practices of young school-aged children, special attention should be paid to female children and those coming from relatively large families.

This is approved by the Amhara Public Health Institute Research Ethics Committee (address: Felege Hiwot Rd, Bahir Dar, Ethiopia; post code: 641). Besides, the data collection have been funded by the Mecha Field and Demographic Health Survey Centre, Bahir Dar Ethiopia (Tis Abay RD, BDU; post code 079). Therefore, accessibility of the data is subject to permission of the above organizations.

**Funding:** The authors received no specific funding for this work.

**Competing interests:** The author declares that there are no competing interests.

## Introduction

Childhood malnutrition including underweight, wasting, and stunting and their consequences are the major global health priority, particularly in low- and middle-income countries [1]. Malnutrition is the most important risk factor for childhood illnesses and deaths globally, with hundreds of millions of young children being affected worldwide [2]. More importantly, the majority of stunting, underweight, wasting and including micronutrient deficiencies in children are occurring in developed regions such as in Asia and sub-Saharan Africa, mainly due to inadequate feeding to meet their growth demand and high burden of infectious diseases in the regions [3, 4].

School age children ranging between 6–12 years has been found to be a critical period of physical, cognitive, and social development of children [5, 6]. However, in case of a problem such as nutrition [7], these vital parameters would not be achieved and may result in chronic malnutrition, intellectual development delay, school failures, and delayed transition to safe adolescent and adulthood [8, 9]. Although, there is lack of comprehensive evidence about the magnitude of malnutrition among school age children, small scale surveys in developing countries indicate that a high burden of chronic malnutrition such as stunting in school age children is observed to be prevalent amongst 40% India [10], 39% in Indonesia [11], and 26% in Bangladesh [12, 13], and 22.5% in Nigeria [14].

Studies in an Ethiopian context have reported a high prevalence of malnutrition among school going children. For example, an study conducted in Addis Ababa [15] and a systematic review [16] reported up to 18% prevalence of underweight among school age children. Also, other small scale surveys reported up to a 57% stunting rate in Humbo located at Southwestern Ethiopia [17], 46% in Northwestern Ethiopia [17], and 42% in southern Ethiopia [18]. Age of child, mother education, parent occupation, family size, food insecurity and poverty were the main predictors of higher prevalence of chronic malnutrition in the regions [17–19]. However, research findings pertaining to the above issues are still inconsistent and inconclusive. While overall, substantial progress has been made in child health programs including school based feeding, nutrition education, awareness creation, periodic deworming, vitamin, a supplementation and vaccinations, the rate of this reduction in the level of malnutrition like stunting has been insufficient to achieve child health-related Sustainable Development Goals (SDGs) of the United Nations [20]. To achieve child health related goals, up-to-date evidence on nutritional status and related factors are needed to evaluate progresses and identify gaps for future intervention.

Using these premises, we conducted this community-based survey to assess nutritional status of school age children by collecting information on socio-demographics, economic status, child diet habits, food insecurity, and others leading to under nutrition.

## Materials and methods

### Study settings and period

A community-based cross-sectional study was conducted between April 1, 2018 and June 15, 2018 in Merawi town, which is located in Amhara regional state in Ethiopia, 535 km away from Addis Ababa, and 30 km from Bahir Dar. Based on the latest projections from the Central Statistical Agency of Ethiopia, Merawi is estimated to have a total population of 35,541. Of these, 18,479 are males and 17,062 are females. Most of the inhabitants (98.91%) practiced Ethiopian Orthodox Christianity. The town has several private and public health clinics, including a public health center, and a public hospital.

The sample size was calculated using a single population proportion formula by considering the following assumptions: P = 50% proportion of malnutrition for school age children, margin of error (d) 5%, and confidence level (CL) 95%, and after considering a 10% non-response rate, the final sample size stood at 422. All three *kebeles* (small administrative units) of the town were included in the study. The number of households in the town was obtained from the administrative offices. We proportionated the sample size based on the number of households in each *kebele*. Households in each *kebele* were then selected using computer generated random numbers. From the selected household, we interviewed mothers with school age children. The youngest child was included in the sample for those with more than one child in a similar age group in the same household. However, we moved to the immediate next household for households without school-age children to conduct our interview.

## Data collection tools and procedure

Data were collected by trained nurses through a structured questionnaire using back translation mechanism. The instrument was translated into Amharic from English, before being translated back and pre-tested for consistency. Data collectors were properly trained in collection techniques and procedures. The data were collected through face-to-face interviews with mothers along with their child (6–12 years of age) to assess socio-demographic variables and environmental characteristics, including maternal/child characteristics, and finally anthropometric measurements.

## Data quality assurance

All research related questionnaires of this study was developed after comprehensive review of the literature and was subsequently pre-tested using 5% of the calculated sample size. Training was given in relation to each module of the questionnaire for data collectors. To ensure data quality, completeness, accuracy, and consistency, all collected data were checked every day by the investigator during the entire data collection period. Any errors related to clarity, ambiguity, incompleteness, and misunderstanding, were resolved on a daily basis.

## Variables of the study

**Dependent variable.** The main outcome variable of the study is the nutritional status (underweight, wasting and stunting) among school age children.

**Independent variable.** A large number of variables were used as independent variables in this study. These include (1) **social and economic variables** such as low food availability, dietary diversity, media access, misconception about certain feedings, inadequate feeding practice during illness, inadequate breastfeeding and weaning practices, late initiation of complementary feeding, access to iodized salts, family planning and number of children, (2) **environmental factors** such as unhygienic living conditions, agricultural, and food shortage; (3) **child and maternal related factors** such as age, sex, birth interval, birth size, breastfeeding, educational status, place of delivery and immunizations. These independent variables are drawn from prior research in similar fields [15, 20–30].

Fig 1 depicts the nexus between dependent and independent variables examined in this research. The research model (Fig 1) represents the category of variables assessed in this study. The diagram on the right-hand side represents the dependent (outcome) variable which is school age children nutritional status. This variable is measured by the level of underweight, wasting and stunting, diet habits and household food security. Weight of the child was measured using a digital weight scale (7506 digital scale) and was recorded in the nearest one decimal point, and height of the child was measured using a fixed non-bending wooden meter.

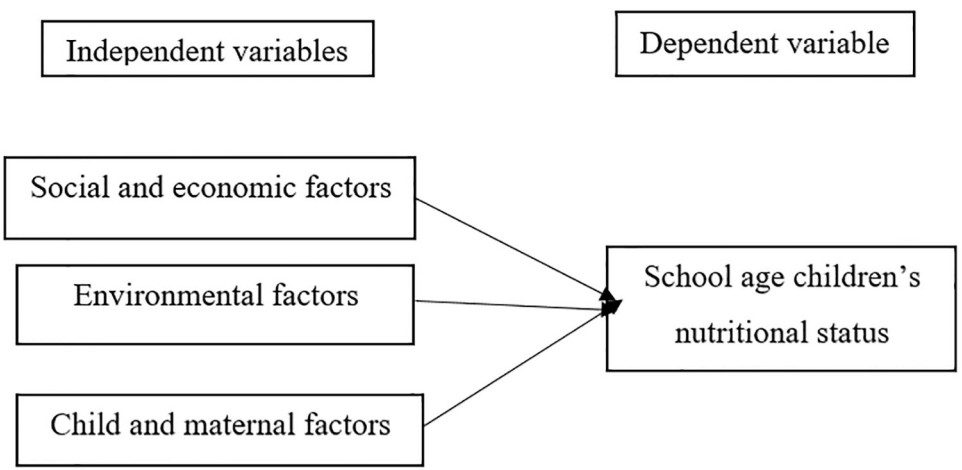

**Fig 1. A research model showing the possible links between the explanatory and outcome variables.**

Children were instructed to take off their shoes to stand upright on their heels, buttocks and shoulders touching the wall. Height of the child was measured and recorded in centimeters. After the measurements were performed, child's nutritional status was defined as being underweight, wasting and stunting, i.e., when the child had one of the three characteristics such that weight for height, height for age, and weight for age less than -2SD from the reference population based on the WHO multicenter growth reference chart 2007 [23].

On the other hand, the three boxes on the left-hand side of the model represents explanatory (independent) variables including socio-economic variables, environmental and child and maternal factors. The social and economic variables which are measured by low foods availability, dietary diversity, media access, misconceptions about certain feedings, inadequate feeding practices during illness, inadequate breastfeeding and weaning practices, late initiation of complementary feeding, access to iodized salts, family planning and number of children. The environmental factors, include sanitation, agriculture (irrigation), and food shortages in the household. Finally, the maternal and child-related factors included age, sex, birth interval, birth size, breastfeeding, educational status, maternal, place of delivery and immunization of the child.

## Data analysis

Collected data were entered and cleaned using Epi data version 3.1 and exported to WHO Anthro Plus and SPSS software version 21 for analysis. Both descriptive and inferential statistics were used to present the data. Binary logistic regression was performed to examine the effect of each independent variable on the outcome variable, and this is presented with adjusted odds ratios (AOR) and 95% confidence intervals (CI). We also examined for the presence of collinearity issue among variables and the results were found to be in acceptable range; variance inflation factor (VIF) was in acceptable rage (VIF< 3). A non-significant Hosmer-Lemeshow model goodness fit *test* was also achieved. Statistical significance was set at the universal cut off point of p = <0.05.

## Ethics approval

Ethical clearance was obtained from the Amhara Public Health Institute. The data collectors informed each parent/guardian and confirmed their willingness to participate by signing an

informed consent sheet. Thus, consent was obtained from each study participant and confidentiality was assured for all the information provided. Moreover, personal identifiers were not included in the questionnaire.

## Results

### Socio-demographic characteristics

Of all eligible participants, 392 constituted a response rate of 92.8%. Just over half (50.8%) of the respondents were males, 47% were between 6–8 years of age, and 30.3% were first in birth order. As far as the age of the mother is concerned, an overwhelming majority (96.5%) were between 18–45 years of age, 59.6% had family members of five and above, 46.5% had 3 to 4 children, and 65.2% were orthodox Christians. Concerning educational status of mothers, 55.8% were high school and above educated, 42.2% were employed, and 82.3% were married. Convresely, 64.7% of husbands were educated at hiigh school level and above. Of these 46.9% were employed, 12.6% had farming land, and 68.7% had an average monthly income of 2001 Ethiopian birr and above. A vast majority of mothers (93.7%) had access to electronic information from different sources (Table 1).

### Maternal and child health related characteristics

Regarding child and maternal health utilization characteristics, 364 (91.9%) had ante-natal follow up, 272 (68.7%) had TT vaccinations, 325 (82.1%) received additional feeding during pregnancy, 361 (91.2%) delivered in health institutions, 393 (99.2%) had experience of breastfeeding, 314 (79.3%) practiced breastfeeding exclusively, and 333 (84.1%) mothers used family planning to control birth.

About 279 (70.5%) children were fully vaccinated, 255 (64.4%) took vitamin A supplement up to five years old, most (86.4%) children were in school with nearly half below grade two. About 55 (13.9%) children engaged in work; amongst them, 20 (36.4%) spent more than three hours per day in work. About 297 (75.0%) had a history of illness: diarrhea (54.5%), pneumonia (31.3%), measles (3%), malaria (2.7%) and others (8.4%). About 171 (43.2%) children had diarrheal morbidity in the past one year, 149 (37.6%) took additional feeding during illness, 382 (96.5%) children had breakfast regularly, and of them 336 (85.1%) had their meals four and more times per day.

A total of 360 (90.9%) mothers had access to child nutrition education, 61 (15.4%) experienced water shortage for cooking, 367 (92.7%) used pipe water, 372 (93.9%) regularly kept child hygiene, 323 (81.6%) regularly washed their hands, 352 (88.9%) cut their nails, 323 (81.6%) used iodized salt, 288 (72.7%) used wood for cooking, and 288 (72.7%) had a modern latrine facility (see Table 2 for details).

### Factors driving malnutrition

Initially variables were tested using bivariate analysis to see their association with the outcome variables (underweight, wasting and stunting). Gender, diarrhea in the past year, age of the child, number of children, husband's education, year of education, and type of salt used were the independent predictors of underweight. Gender, age of the child, farming land, irrigation, diarrheal illness, breakfast, and mother education about child feeding were the independent predictors of wasting. Vaccination, Vitamin A supplement, child work engagement, age of the child, mother's age, child's level of education, and food served at work were the independent predictors of stunting. Age of the child and number of children were the final predictors for being underweight, while it was gender for wasting, and age of the child for stunting.

**Table 1. Socio-demographic distribution of school age children in Merawi town, Northwest, Ethiopia, 2018 (n = 396).**

| Variable | Response | Total n (%) | Underweight n (%) | Stunted n (%) | Wasted n (%) |
|---|---|---|---|---|---|
| Sex of the child | Male | 201 (50.8) | 12 (3.8) | 17(4.3) | 28 (7.1) |
| | Female | 195 (49.2) | 6 (1.9) | 22 (5.6) | 13 (3.3) |
| Age of the child | 6–8 years | 186 (47.0) | 4 (1.3) | 3(0.8) | 25 (6.3) |
| | 9–10 years | 134 (33.8) | 13 (4.1) | 18 (4.5) | 14(3.5) |
| | 11–12 years | 76 (19.2) | 1(0.3) | 18 (4.5) | 2 (0.5) |
| Number of children | 1–2 | 179 (45.2) | 10 (3.1) | 18 (4.5) | 20 (5.1) |
| | 3–4 | 184 (46.5) | 5 (1.6) | 17 (4.3) | 20 (5.1) |
| | Five and above | 33 (8.3) | 3(0.9) | 4(1.0) | 1(0.3) |
| Number of family | 1–2 | 10 (2.5) | 10 (3.1) | 1(0.3) | 1(0.3) |
| | 3–4 | 150 (37.9) | 5(1.6) | 16 (4.0) | 16(4.0) |
| | 5+ | 236 (59.6) | 3 (0.9) | 22(5.6) | 22 (5.6) |
| Birth order of the child | First | 120 (30.3) | 5 (1.6) | 11 (2.8) | 12 (3.0) |
| | Second and above | 276 (69.7) | 13 (4.1) | 28 (7.1) | 29 (7.3) |
| Age of mother | 18–45 | 381 (96.2) | 18 (5.7) | 34 (8.6) | 41 (10.4) |
| | 46–60 | 15 (3.8) | 0 (0.0) | 5 (1.3) | 0 (0.0) |
| Religion | Orthodox | 258 (65.2) | 8 (2.5) | 21 (5.3) | 23(5.8) |
| | Muslim | 100 (25.3) | 9(2.8) | 16(4.0) | 13 (3.3) |
| | Protestant | 38 (9.6) | 1 (0.3) | 2(0.5) | 5(1.3) |
| Mother education | Educated | 221 (55.8) | 10(3.1) | 23 (5.8) | 23(5.8) |
| | Uneducated | 175 (44.2) | 8 (2.5) | 16 (4.0) | 18 (4.5) |
| Occupation of mother | Employed | 167 (42.2) | 7 (2.2) | 19 (4.8) | 17 (4.3) |
| | Farmer | 13 (3.3) | 0 (0.0) | 1(0.3) | 0 (0.0) |
| | Unemployed | 216 (54.5) | 11 (3.5) | 19 (4.8) | 24 (6.1) |
| Marital status | Married | 326 (82.3) | 16 (5.0) | 30 (7.6) | 34 (8.6) |
| | Unmarried | 70 (17.7) | 2 (0.6) | 9 (2.3) | 7 (1.8) |
| Husband level education | Educated | 233 (64.7) | 9 (2.8) | 26(6.6) | 26 (6.6) |
| | Uneducated | 127 (35.3) | 9 (2.8) | 12 (3.1) | 15 (3.8) |
| husband's occupation | Employed | 169 (46.9) | 7(2.2) | 18 (4.6) | 20(5.1) |
| | Unemployed | 191 (53.1) | 11 (3.6) | 20(4.6) | 20 (5.2) |
| Farming land | Yes | 50 (12.6) | 2 (0.6) | 3(0.8) | 2 (0.5) |
| | No | 346 (87.4) | 16 (5.0) | 36 (9.1) | 39 (9.8) |
| Irrigation user | Yes | 30 (7.6) | 1(0.3) | 2 (0.5) | 1 (0.3) |
| | No | 366 (92.4) | 17 (5.3) | 37 (9.3) | 40 (10.1) |
| Average monthly income | < 1000 | 40 (10.1) | 1 (0.3) | 3 (0.8) | 2 (0.5) |
| | 1001–2000 | 84 (21.2) | 5 (1.6) | 7 (1.8) | 8 (2.0) |
| | > 2001 | 272 (68.7) | 12 (3.8) | 29 (7.3) | 31 (7.8) |
| A radio and or television | Yes | 370 (93.4) | 17 (5.3) | 39 (9.8) | 39 (9.8) |
| | No | 26 (6.6) | 1(0.3) | 0(0.0) | 2 (0.5) |

After identifying variables in the bivariate analysis, we performed multivariate logistic regression analysis to determine the significant variables that determined nutritional status of school ages in the study area. Thus, in the binary logistic regression model, the age of the child emerged as the key determinant for being underweight. A child between 6–8 years of age has an almost 13 times higher chance of having a normal weight than a child of 9–12 years of age [AOR = 12.930 (2.350, 71.157)]. On the other hand, the number of children emerged as the determinant for being underweight. Children living within families with 3–4 children were almost eight times more likely to have a normal weight than those from families with five or

**Table 2.  Maternal related characteristics distribution of school age children in Merawi town, Northwest, Ethiopia, 2018 (n = 396).**

| Variables | Response | Frequency | Percent |
|---|---|---|---|
| Antenatal care follows up | Yes | 364 | 91.9 |
| | No | 32 | 8.1 |
| TT Immunization status | Completed | 272 | 68.7 |
| | Incomplete | 101 | 25.5 |
| | Not vaccinated | 23 | 5.8 |
| Additional feeding during pregnancy | Yes | 325 | 82.1 |
| | No | 71 | 17.9 |
| Place of delivery | Health facility | 361 | 91.2 |
| | Home | 35 | 8.8 |
| History of breastfeeding | Yes | 393 | 99.2 |
| | No | 3 | 0.8 |
| EBF | Yes | 314 | 79.3 |
| | No | 82 | 20.7 |
| Vaccination status | Completed | 279 | 70.5 |
| | Not completed | 109 | 27.5 |
| | Not vaccinated | 8 | 2.0 |
| Vitamin A supplementation | Yes | 255 | 64.4 |
| | Not completed | 128 | 32.3 |
| | No | 13 | 3.3 |
| Ever used of FP methods | Yes | 333 | 84.1 |
| | No | 63 | 15.9 |
| Is your child on school | Yes | 342 | 86.4 |
| | No | 54 | 13.6 |
| Level/ grade of education | 1–2 grade | 172 | 49.7 |
| | Grade three above | 174 | 50.3 |
| Is your child engaged in work | Yes | 55 | 13.9 |
| | No | 341 | 86.1 |
| If engaged in work for how many hours | 1–3 hours | 31 | 55.4 |
| | Above three hours | 25 | 44.6 |
| How can the child get feeding if he engages in work for more than 3 hours | Yes | 20 | 36.4 |
| | No | 35 | 63.6 |
| Child history of illness | Yes | 297 | 75.0 |
| | No | 99 | 25.0 |
| Type of illness | Pneumonia | 93 | 31.3 |
| | Diarrhea | 162 | 54.5 |
| | Measles | 9 | 3.0 |
| | Malaria | 8 | 2.7 |
| | Others /specify | 25 | 8.4 |
| Child diarrheal morbidity in the past year one year | Yes | 171 | 43.2 |
| | No | 225 | 56.8 |
| Child diarrheal morbidity in the last two weeks | Yes | 27 | 6.8 |
| | No | 369 | 93.2 |
| Type feeding do you give for your child during illnesses | Regular family dish | 247 | 62.4 |
| | Additional feeding | 149 | 37.6 |
| Do the child eat breakfast regularly | Yes | 382 | 96.5 |
| | No | 14 | 3.5 |
| Frequency of eating per day | Four and above | 336 | 85.1 |
| | 1–3 | 59 | 14.9 |

(*Continued*)

**Table 2.** (Continued)

| Variables | Response | Frequency | Percent |
|---|---|---|---|
| Access to child nutrition education | Yes | 360 | 90.9 |
| | No | 36 | 9.1 |
| Water shortage for cooking | Yes | 61 | 15.4 |
| | No | 335 | 84.6 |
| Type of water used for cooking | Pipe | 367 | 92.7 |
| | Dam water River /follow water | 29 | 7.3 |
| Boiling water for serving a child | Yes | 22 | 5.6 |
| | No | 374 | 94.4 |
| Do you regularly keep your child hygiene | Yes | 372 | 93.9 |
| | No /pipe | 24 | 6.1 |
| Regular hand washing | Yes | 323 | 81.6 |
| | No | 73 | 18.4 |
| Nail cutting | Yes | 352 | 88.9 |
| | Only mine | 35 | 8.8 |
| | No | 9 | 2.3 |
| Type of salt used for cooking | Iodized | 323 | 81.6 |
| | Normal/dish salt | 73 | 18.4 |
| Type cooking fuel | Wood | 288 | 72.7 |
| | Petroleum gas | 13 | 3.3 |
| | electricity | 95 | 24.0 |
| Type of toilet | Modern latrine | 288 | 72.7 |
| | Temporary toilet/ open | 108 | 27.3 |

**Keys**: Regular breakfast: when a child always eat breakfast; Frequency of eating per day: number of meals per day; Water shortage for cooking: when there is lack of access to clean water like tap water for cooking.

more children [AOR = 8.155 (1.312, 50.677)] (see Table 3). By contrast, wasting was affected by the gender of the child. A male child had a 55% lower chance to have a normal body mass index for his age than those of female children [AOR = 0.455(0.224, 0.927)] (see Table 4). This study revealed age of the child as the driving factor of height for age (stunting). Children within the age group of 6–8 years were almost 13 times more likely to be affected by stunting than those of children within the age group of 9–12 years [AOR = 12.369 (2.522, 60.656)] (Table 5).

## Discussion

Prevalence of being underweight stood at 5.7% in this research. This result is much lower than that of the prior studies conducted: 16% in Adds Ababa [15], 28% in rural Northwest Ethiopia [31], and from neighboring country Kenya (14.9%) [28]. This could be attributed to the fact that in the study area, child eating habits were extended to almost three times per day, while fasting occurred at the early age group, and there was an absence of school-based feeding.

This research revealed the age of the child to be the determinant factor for being underweight. A child between 6–8 years of age is almost 13 times more likely to have a normal weight for their age than that of a 9–12-year-old child. On the other hand, the number of children in a family was the determinant for being underweight. Children living with 3–4 children in a family were almost eight times more likely to have normal weight than those from families

**Table 3. Factors associated with underweight school age children in Merawi town, Northwest, Ethiopia, 2018 (n = 396).**

| Variables | Weight for age | | | | | |
|---|---|---|---|---|---|---|
| | Response | Normal | Under weight | COR (95% CL) | AOR (95% CL) | P- value |
| Sex | Male | 151 | 12 | 0.507 (0.185, 1.385) | | |
| | Female | 149 | 6 | 1 | | |
| Diarrhea last year | Yes | 128 | 5 | 1.935 (0.673, 5.565) | | |
| | No | 172 | 13 | 1 | | |
| Age of the child | 6–8 | 180 | 4 | 5.250 (1.688, 16.333) | **12.930 (2.350, 71.157)** | **0.003** |
| | 9–12 | 120 | 14 | 1 | 1 | |
| No of children | 1–2 | 146 | 10 | 2.737 (0.682, 10.986) | | |
| | 3–4 | 138 | 5 | 5.175 (1.129, 23.710) | **8.155 (1.312, 50.677)** | **0.024** |
| | 5+ | 16 | 3 | 1 | | |
| Husband education | Educated | 193 | 9 | 1.838 (0.708, 4.772) | | |
| | Uneducated | 105 | 9 | 1 | | |
| Grade/ level of education | 1–2 | 153 | 6 | 2.383 (0.841,6.755) | | |
| | 3+ | 107 | 10 | 1 | | |
| Type of salt | Iodized | 242 | 17 | 0.245 (0.032,1.882) | | |
| | Non iodized | 58 | 1 | 1 | | |

Keys: COR: Crude Odds ratio; AOR: Adjusted Odds Ratio; N: Number

with five or more children. This finding is consistent with the research findings in Addis Ababa [15], a systematic review in Ethiopia [16] and in Nairobi [28], and may be attributed to the fact that when the child is at a young school age there is a risk of poor appetite in relation to the school environment, and when the number of children is increased it may create acute food shortages in poor families resulting in negligent child-rearing habits.

The prevalence of wasting was 9.8%, which was similar with 9% in Gonder [17] but lower than the that of the findings and other studies in Uganda [32], in Afghanistan [33] and in Kenya [28]. This may be due to accelerated growth in this age group, which may be related to

**Table 4. Distribution of body mass index for age among urban school age children in Merawi, Northwest Ethiopia, 2018 (n = 396) (urban).**

| Variables | Body mass index for age | | | | | |
|---|---|---|---|---|---|---|
| | Response | Normal | Low | COR (95% CL) | AOR (95% CL) | P- value |
| Sex | Male | 173 | 28 | 0.441(0.221, .880) | **0.455(.224, .927)** | **0.030** |
| | Female | 182 | 13 | 1 | 1 | |
| Age of child | 6–8 | 161 | 25 | 0.531(0.274, 1.029) | | |
| | 9–12 | 194 | 16 | 1 | | |
| Farming land | Yes | 48 | 2 | 3.049 (0.713, 13.039) | | |
| | No | 307 | 39 | 1 | | |
| Irrigation | Yes | 29 | 1 | 3.558(0.472, 26.832) | | |
| | No | 326 | 40 | 1 | | |
| Diarrheal illness | Yes | 158 | 13 | 1.727(0.866, 3.445) | | |
| | No | 197 | 28 | 1 | | |
| Breakfast | Yes | 344 | 38 | 2.469 (0.660, 9.241) | | |
| | No | 11 | 3 | 1 | | |
| Education about child feeding | Yes | 320 | 40 | 0.229(0.030, 1.714) | | |
| | No | 35 | 1 | 1 | | |

**Table 5. Distribution of Height for age among urban school age children in Merawi, Northwest Ethiopia, 2018 (n = 396).**

| Variables | Height for age | | | | | |
|---|---|---|---|---|---|---|
| | Responses | Normal | Stunted | COR (95% CL) | AOR (95% CL) | P- value |
| Vaccination | Yes | 258 | 21 | 2.234(1.142, 4.369) | | |
| | No | 99 | 18 | 1 | | |
| Vitamin A vaccination | Yes | 238 | 17 | 2.588(1.324, 5.058) | | |
| | No | 119 | 22 | 1 | | |
| Child work engagement | Yes | 45 | 10 | 0.418(0.191, 0.916) | | |
| | No | 312 | 29 | 1 | | |
| Age of the child | 6–8 | 183 | 3 | 12.621(3.817,41.732) | **12.369 (2.522, 60.656)** | **0.002** |
| | 9–12 | 174 | 36 | 1 | | |
| Mother age | 20–45 | 347 | 34 | 5.103(1.649,15.794) | | |
| | 46+ | 10 | 5 | 1 | | |
| Child level of education | 1–2 | 165 | 7 | 4.141(1.746, 9.821) | | |
| | 3+ | 148 | 26 | 1 | | |
| Food serving on work | Normal | 226 | 21 | 1.479(.760,2.876) | | |
| | Additional | 131 | 18 | 1 | | |

good feeding, in terms of access to food and diversity of food. By contrast, wasting was affected by the sex of the child. A male child had a 55% lower chance of having a normal body mass index for his age, which is comparable to the findings in Maputo, Mozambique [34], in Western Kenya [35], Nairobi [28], and India [25]. This finding points to a subconscious practice to pay preferential attention to male children and this practice is prevalent in many underdeveloped countries in Asia and Africa. In this study the prevalence of stunting was found to be at 10.4% which is lower than those study findings in 16% in Adds Ababa [15], 42% Southern Ethiopia [18], 46% in Gonder [17], and from Uganda of 23.8% [32], in Kenya of 30.2% [28], and in Abeokuta, Southwest Nigeria of 17.4% [36]. This could be due to the socio-cultural and economic differences between countries from the referenced areas. In this study, the age of the child emerged as the determinant factor of height for age or stunting. Children within the age group of 6–8 years were almost 13 times more likely to be affected by stunting than those within the age group of 9–12 years. This finding corroborates the research finding in relation to Afghanistan [33, 37], India [25], in Gonder [33] and in Addis Ababa, Ethiopia [15]. This may be due to very young school age children being at risk of adopting school-based feeding, the distance of the school from the home, and a change of the environment at school from home.

The findings of the study make significant contributions to knowledge in relation to the critical roles played by age, number of children in a family, and the gender of the child in driving nutritional status among children in a developing country such as Ethiopia. In particular, this research has contributed to the realm of our knowledge gap in terms of feeding practices of young school-aged children especially female children often receiving comparatively less attention as well as children belonging to large families.

## Study limitations and potential for further study

Despite its unique contributions, this study has a number of limitations. First, this is a community-based study using information provided by mothers about their children to assess the nutritional status of their school age children, which might have information recall bias to

some variables related to time. Second, estimates of a child's nutrition were measured using the internationally used standard tools and definitions, which did not consider country specific parameters. This might underestimate or overestimate the local situation. Third, this research only used a structured interviewer-administered questionnaire and measurements that might not reflect culture and perception of food taboos in the community. Fourth, this research was confined to a particular region of Ethiopia with a relatively small sample size which might affect the generalizability of the findings across the borders in the country and beyond. However, future research could use the result to conduct more broad-based sample drawn from several regions in Ethiopia and beyond to come up with better findings for practice and research about child nutrition in developing regions.

## Concluding remarks

This research addressed some critical issues in the public health arena that were not well covered in the existing literature, especially in the context of a developing African country, Ethiopia where malnutrition is highly prevalent in school aged children. In particular, this study identified the possible determinants of underweight, wasting and stunting among school aged children (6–12 years). The outcomes of this study have an important bearing on designing appropriate policies to address the current problems pertaining to the above issues examined in this research. One particular issue emerging from this research that warrants the attention of policymakers is that female children receive lesser attention than their male counterparts in terms of feeding. Furthermore, attention can be directed to children belonging to large families as they are more likely to suffer from malnutrition and are underweight for an obvious reason. Age of children is another issue deserving attention because malnutrition leading to underweight is more common in older students than their younger counterparts. Although the data of this study was obtained from mothers and might involve recall bias to some variable which suggests the importance of prospective and rigorous to investigate causal associations, this study shows that school age children feeding practices need to be improved and adjusted to the children cohorts identified in this research. Finally, since, healthy work force is vital for accelerating the pace of economic growth of a transitional economy such as Ethiopia, policymakers should address the above-mentioned issues as an early intervention strategy towards developing healthy human resources for Ethiopia to reap the benefits of demographic dividend.

## Author Contributions

**Conceptualization:** Tilahun Tewabe.

**Data curation:** Tilahun Tewabe, Md. Moustafa Kamal.

**Formal analysis:** Tilahun Tewabe, Md. Moustafa Kamal, Khorshed Alam, Ali Quazi, Majharul Talukder, Syeda Z. Hossain.

**Funding acquisition:** Tilahun Tewabe.

**Investigation:** Tilahun Tewabe.

**Methodology:** Tilahun Tewabe, Md. Moustafa Kamal, Khorshed Alam, Ali Quazi, Majharul Talukder, Syeda Z. Hossain.

**Project administration:** Tilahun Tewabe.

**Software:** Tilahun Tewabe, Md. Moustafa Kamal.

**Writing – original draft:** Tilahun Tewabe, Md. Moustafa Kamal, Khorshed Alam, Ali Quazi, Majharul Talukder, Syeda Z. Hossain.

**Writing – review & editing:** Tilahun Tewabe, Md. Moustafa Kamal, Khorshed Alam, Ali Quazi, Majharul Talukder, Syeda Z. Hossain.

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
