## [Decision Letter · Decision Letter 0]

13 Apr 2022

PGPH-D-22-00367

Factors driving underweight, wasting and stunting among urban school aged children: Evidence from Ethiopia

Dear Dr. Kamal,

Thank you for submitting your manuscript to PLOS Global Public Health. After careful consideration, we feel that it has merit but does not fully meet PLOS Global Public Health’s publication criteria as it currently stands. Therefore, we invite you to submit a revised version of the manuscript that addresses the points raised during the review process.

Thank you for submitting your manuscript to PGPH for publication. It has now been reviewed by two independent reviewers. Although they felt that it has some merit, they have raised major issues, which you must address before your paper can be considered for publication. The issues centre around lack of clarity in your presentation, as well as inconsistencies in many areas. There is also the need for you to revisit your statistical analysis to ensure that it is rigorous enough.  

One of the reviewers suggested that it is inappropriate to use child's age as a predictor of stunting and underweight, because these indicators were computed, taking into account the child's age. This is inaccurate. Age is a key covariate of childhood underweight and stunting. Therefore, you do not need to remove the child's age from the regression models. However, you need to explain why you think age is an important covariate/predictor in your rebuttal letter. 

Please ensure that your decision is justified on PLOS Global Public Health’s publication criteria and not, for example, on novelty or perceived impact.

We look forward to receiving your revised manuscript.

Kind regards,

Dickson Abanimi Amugsi, PhD

Academic Editor

Journal Requirements:

1. Your co-authors, Tilahun Tewabe (tilahun.alamnia@anu.edu.au), Ali Quazi (Ali.Quazi@canberra.edu.au), Majharul Talukder (Majharul.Talukder@canberra.edu.au), and Syeda Z Hossain (zakia.hossain@sydney.edu.au), have not confirmed authorship of the manuscript. We have resent them the authorship confirmation email; however please check that the above email address for them is correct and follow up personally to ensure they confirm. Please note that we cannot pass your manuscript to Production until we have received confirmations from all co-authors.

Just in case your co-authors are having difficulty confirming their authorship, you may advise them to send us an email at globalpubhealth@plos.org and we will confirm their authorship on the authors' behalf.

2. Please amend your Financial Disclosure statement. If you did not receive any funding for this study, please simply state: “The authors received no specific funding for this work.”

3. Please update your Competing Interests statement. If you have no competing interests to declare, please state: “The authors have declared that no competing interests exist.”

4. Please provide a complete Data Availability Statement in the submission form, ensuring you include all necessary access information or a reason for why you are unable to make your data freely accessible. Note that it is not acceptable for the authors to be the sole named individuals responsible for ensuring data access.

PLOS defines a study's minimal data set as the underlying data used to reach the conclusions drawn in the manuscript and any additional data required to replicate the reported study findings in their entirety. Any potentially identifying patient information must be fully anonymized. 

If your research concerns only data provided within your submission, please write “All data are in the manuscript and/or supporting information files"" as your Data Availability Statement.”

5. Please provide separate figure files in .tif or .eps format only and remove any figures embedded in your manuscript file. Please ensure that all files are under our size limit of 20MB.

Additional Editor Comments (if provided):

Thank you for submitting your manuscript to PGPH for publication. It has now been reviewed by two independent reviewers. Although they felt that it has some merit, they have raised major issues, which you must address before your paper can be considered for publication. The issues centre around lack of clarity in your presentation, as well as inconsistencies in many areas. There is also the need for you to revisit your statistical analysis to ensure that it is rigorous enough.

One of the reviewers suggested that it is inappropriate to use child's age as a predictor of stunting and underweight, because these indicators were computed, taking into account the child's age. This is inaccurate. Age is a key covariate of childhood underweight and stunting. Therefore, you do not need to remove the child's age from the regression models. However, you need to explain why you think age is an important covariate/predictor in your rebuttal letter.

Reviewers' comments:

Reviewer's Responses to Questions

**Comments to the Author**

1. Does this manuscript meet PLOS Global Public Health’s publication criteria? Is the manuscript technically sound, and do the data support the conclusions? The manuscript must describe methodologically and ethically rigorous research with conclusions that are appropriately drawn based on the data presented.

Reviewer #1: Partly

Reviewer #2: Yes

2. Has the statistical analysis been performed appropriately and rigorously?

Reviewer #1: No

Reviewer #2: No

3. Have the authors made all data underlying the findings in their manuscript fully available (please refer to the Data Availability Statement at the start of the manuscript PDF file)?

Reviewer #1: Yes

Reviewer #2: No

4. Is the manuscript presented in an intelligible fashion and written in standard English?

Reviewer #1: Yes

Reviewer #2: No

5. Review Comments to the Author

Reviewer #1: Thank you very much for partaking in this essential work. This cross-sectional study probed factors driving underweight, wasting, and stunting among urban school-aged children: Evidence from Ethiopia. The issue is still requiring attention and the findings are remarkable. However, I have a few concerns that could be clarified and amended for better understanding.

Comments and questions:

• On the abstract part:

Background - line 7 and result- line 3, the word “determinants”, is better replaced by another suitable action verb, since you have conducted a cross-sectional study. Similarly, on the same line, the phrase “…identify and assess…” should be rearranged.

The method of sampling should be specified as well.

• On the background part:

The first paragraph reference number 1 and 4 talk about under-five children instead of your focus area which is school-age children.

In the second paragraph, there is a mix of ideas i.e. initially you stated global magnitude of malnutrition (first line), then about SAM (second line), consequently about risk factors (from line 3 to 5), finally global magnitude of malnutrition again (from line 6 to 8). Therefore, it is better to develop paragraphs with one topic sentence and other supporting sentences separately.

The 3rd paragraph – reference number 8 does not match on the list of references (EDHS 2011 instead of EDHS 2016). And these children are under-five as well.

The 3rd paragraph – lines number 4 and 5 (… 700,000 pregnant and breastfeeding women …) is out of the scope of this study.

The 3rd paragraph – line number 6 (… the rate of this reduction in under-five …) better to focus on school-aged children.

• Methods and materials

Variables of the study

You said your dependent variable is the nutritional status (underweight, wasting, and stunting) among under-five children. Why?

On independent variables line number 6 (environmental factors such as nhygienic …) requires edition. Additionally, what do you mean by “birth size” on line number 7? And what is its relevance, since it is not included in your result part?

On page 6 the 1st paragraph line number 4 – you stated that “the variable (nutritional status) is measured by the level of underweight, wasting and stunting, diet habits and household food security.” How did you measure and operationalize household food security)? And it is not included in your result part. Thus, it is better to remove diet habits and household food security.

On page 6 the 2nd paragraph, it is already described on page 5 under independent variables. Therefore, what is the value of repeating it?

Data analysis, result, and discussion

First paragraph lines number 3 and 4 – do you mean AORs are used for presenting multivariable logistic regression analysis output?

Socio-demographic characteristics

On page 7, 392 response rate is not consistent with the one indicated throughout the tables (396).

Factors driving malnutrition

Before identifying factors driving malnutrition, you have to put the nutritional status of school-age children in terms of underweight, stunting, and wasting.

Paragraph one line one – you said that “… bivariate analysis…” , do you mean bivariable logistic regression analyses?

The dependent variable underweight (weight for age) is the composite measure of malnutrition. But, the age of children is considered a predictor of being underweight, which is inappropriate. Therefore, the analysis has to be done again.

Similarly, since stunting is measured using height for age, using age as an independent variable is inappropriate. Therefore, the analysis has to be done again.

Table 3, 4, and 5 – all of the confidence intervals constructed are wide. This is related to using small sample size, hence to come up with better analysis output, categories with very small values shall be revised and merged.

• Discussion

Paragraph one, lines 4, 5, and 6 – “This could be attributed to the fact that in the study area, child eating habits were extended to almost three times per day, …”. Have you come across other areas children’s feeding habits to say this? “ … while fasting occurred in the early age group, and there was an absence of school-based feeding.” Do you think these ideas support your argument for having lower underweight compared with the aforementioned areas?

Third paragraph, line 2 – What about the timing of the research they conducted? For example, reference numbers 16 (2008) and 19(1998). It is better to use recent references for a reasonable comparison. This comment is also applied to the prevalence of stunting since similar references are used (Paragraph three, line 10).

• Concluding remarks

Line one – “… arena …”, do you mean area?

Nevertheless, the above comments and questions are provided, the paper’s publishability depends on the authors’ reactions to revise their manuscript commented and response for review questions accordingly.

Reviewer #2: Feedback to authors

The grammatical and sentence structure needs modification as general comments.

1. Abstract section

Results: ………….. key determinants of underweight, and the gender of the child was significantly associated with wasting [AOR= 0.455 (0.224, 0.927)]. The word determinants are better replaced by associated factors since your study design was cross-sectional.

Conclusion: make clear the possible early interventions that you intend

2. Background

Take updated data about the prevalence of nutritional status with updated references worldwide

Take the Ethiopian mini-EDHS 2019 data

Explain the interventions regarding the Ethiopian context of school health and nutrition initiatives

3. Methods and materials

During your data quality assurance of questionnaire, you mentioned that as you pre-tested in 5% of the calculated sample size what was your chrome batch alpha?

During your data analysis specify your cut of the p-value for eligibility to be screened for the multivariate logistic regression model

Ethical approval and consent to participate letter date and number given by should be mentioned

4. Result

Maternal and child health-related characteristics variables in Table 2 need operational definitions of how they were measured:

A total of 360 (90.9%) mothers had access to child nutrition education,

(255 (64.4 %) took vitamin A supplement up)

55 (13.9%) children engaged in work,

61 (15.4%) experienced water shortage for cooking,

367 (92.7%) used pipe water,

372 (93.9%) regularly kept child hygiene

323 (81.6%) regularly washed their hands,

352 (88.9%) cut their nails,

323 (81.6%) used iodized salt,

288 (72.7%) had a modern latrine facility

NB :

Accessibility of child nutrition education how it was measured?

vitamin A supplementation: from the standard of supplementation how many times has the child has supplemented from his/her age

modern latrine facility: what were the criteria used to determine a latrine to be said modern? The same is true for the above-mentioned factors

5. Discussion

Give evidence-based expiation for gender and size variables

6. Conclusion

Indicate specifically the possible interventions for concerned body based on your findings.

6. PLOS authors have the option to publish the peer review history of their article (what does this mean?). If published, this will include your full peer review and any attached files.

**Do you want your identity to be public for this peer review?** For information about this choice, including consent withdrawal, please see our Privacy Policy.

Reviewer #1: **Yes: **Yirdaw Melese Yilma

Reviewer #2: No

---

## [Decision Letter · Decision Letter 1]

27 Jul 2022

PGPH-D-22-00367R1

Factors driving underweight, wasting and stunting among urban school aged children: Evidence from Ethiopia

Dear Mr. Kamal,

Thank you for submitting your manuscript to PLOS Global Public Health. After careful consideration, we feel that it has merit but does not fully meet PLOS Global Public Health’s publication criteria as it currently stands. Therefore, we invite you to submit a revised version of the manuscript that addresses the points raised during the review process.

Thank you for choosing PGPH. Two additional independent reviewers have assessed your paper and found it to have merit, but there are substantial weaknesses associated with the manuscript that need to be addressed to make it publishable. While Reviewer 4 raised substantial issues regarding the data analysis, Reviewer 3 raised major issues in every aspect of the manuscript.

Although, the earlier two independent reviewers recommended that the manuscript should be accepted for publication, I identified major issues after assessing it. Hence, my decision not to accept their recommendation. Indeed, the current reviewers have highlighted most of the concerns I had with your manuscript. I suggest you adequately address their comments and ensure the quality of the paper is improved...you may seek support from senior colleagues where necessary.

We look forward to receiving your revised manuscript.

Kind regards,

Dickson Abanimi Amugsi, PhD

Academic Editor

Journal Requirements:

1. Please include a separate legend for Figure 1 in your manuscript.

Additional Editor Comments (if provided):

Reviewers' comments:

Reviewer's Responses to Questions

**Comments to the Author**

1. If the authors have adequately addressed your comments raised in a previous round of review and you feel that this manuscript is now acceptable for publication, you may indicate that here to bypass the “Comments to the Author” section, enter your conflict of interest statement in the “Confidential to Editor” section, and submit your "Accept" recommendation.

Reviewer #1: All comments have been addressed

Reviewer #2: All comments have been addressed

Reviewer #3: (No Response)

Reviewer #4: (No Response)

2. Does this manuscript meet PLOS Global Public Health’s publication criteria? Is the manuscript technically sound, and do the data support the conclusions? The manuscript must describe methodologically and ethically rigorous research with conclusions that are appropriately drawn based on the data presented.

Reviewer #1: Yes

Reviewer #2: Yes

Reviewer #3: Partly

Reviewer #4: Yes

3. Has the statistical analysis been performed appropriately and rigorously?

Reviewer #1: Yes

Reviewer #2: Yes

Reviewer #3: No

Reviewer #4: Yes

4. Have the authors made all data underlying the findings in their manuscript fully available (please refer to the Data Availability Statement at the start of the manuscript PDF file)?

Reviewer #1: Yes

Reviewer #2: Yes

Reviewer #3: Yes

Reviewer #4: No

5. Is the manuscript presented in an intelligible fashion and written in standard English?

Reviewer #1: Yes

Reviewer #2: Yes

Reviewer #3: No

Reviewer #4: Yes

6. Review Comments to the Author

Reviewer #1: Dear researchers, thank you very much for addressing and responding to the comments and questions forwarded.

Reviewer #2: Good all my comments have been addressed.

Reviewer #3: (No Response)

Reviewer #4: (No Response)

7. PLOS authors have the option to publish the peer review history of their article (what does this mean?). If published, this will include your full peer review and any attached files.

**Do you want your identity to be public for this peer review?** For information about this choice, including consent withdrawal, please see our Privacy Policy.

Reviewer #1: No

Reviewer #2: No

Reviewer #3: No

Reviewer #4: No

---

## [Decision Letter · Decision Letter 2]

12 Dec 2022

PGPH-D-22-00367R2

Factors driving underweight, wasting, and stunting among urban school aged children: Evidence from Merawi town, Northwest Ethiopia.

Dear Dr. KAMAL,

Thank you for submitting your manuscript to PLOS Global Public Health. After careful consideration, we feel that it has merit but does not fully meet PLOS Global Public Health’s publication criteria as it currently stands. Therefore, we invite you to submit a revised version of the manuscript that addresses the points raised during the review process.

We look forward to receiving your revised manuscript.

Kind regards,

Jitendra Kumar Singh, PhD

Academic Editor

Journal Requirements:

Additional Editor Comments (if provided):

This is overall interesting paper that covers an area of great interest. The authors have made a very good effort overall to describe their findings, considering this is a topic highly quantitative. There are however some areas they further need to address.

Reliability and validity are incomplete. The authors should elaborate on that section.

Reviewers' comments:

Reviewer's Responses to Questions

**Comments to the Author**

1. If the authors have adequately addressed your comments raised in a previous round of review and you feel that this manuscript is now acceptable for publication, you may indicate that here to bypass the “Comments to the Author” section, enter your conflict of interest statement in the “Confidential to Editor” section, and submit your "Accept" recommendation.

Reviewer #4: (No Response)

2. Does this manuscript meet PLOS Global Public Health’s publication criteria? Is the manuscript technically sound, and do the data support the conclusions? The manuscript must describe methodologically and ethically rigorous research with conclusions that are appropriately drawn based on the data presented.

Reviewer #4: Yes

3. Has the statistical analysis been performed appropriately and rigorously?

Reviewer #4: Yes

4. Have the authors made all data underlying the findings in their manuscript fully available (please refer to the Data Availability Statement at the start of the manuscript PDF file)?

Reviewer #4: No

5. Is the manuscript presented in an intelligible fashion and written in standard English?

Reviewer #4: Yes

6. Review Comments to the Author

Reviewer #4: The revision done by the authors helped improve the quality of the manuscript. However, the authors failed to provide a scientific reason for not adjusting for the 3 kebeles in their analysis, an issue I raised earlier. Their response that “However, the city has only three kebeles (small administrative units) only used for administrative purposes and the population among the kebeles are homogeneous, there is no difference among them” cannot be supported with the available data in the manuscript, and this must be proven scientifically and presented in the manuscript. Per their response, the authors are saying that the nutritional outcomes of children from the 3 different Kebeles is the same BUT they failed to provide results to support this claim in the manuscript. They should include the kebele as a predictor/covariate in their model.

In the Abstract and Discussion sections, the authors stated that “prevalence of being underweight, wasting and stunting was found to be 5.7%, 9.8%, 10.4%, respectively”. However, in the ‘Author summary’ section, the authors again stated that “The magnitude of underweight, wasting, stunting was 5.8 %, 10.8%, and 11.6%, respectively” which are completely different. How were the magnitude and the prevalence estimated for which reason the authors were reporting different values? This should be clarified or corrected.

Also, there are still some typo errors that the authors should correct. For example, ‘under weighs’ in line 235 at page 9. Also, there should be dot (.) before ‘Gender’ in line 235 at page 9. Similar typo error in line 206 at page 11 for ‘motheris’.

7. PLOS authors have the option to publish the peer review history of their article (what does this mean?). If published, this will include your full peer review and any attached files.

**Do you want your identity to be public for this peer review?** For information about this choice, including consent withdrawal, please see our Privacy Policy.

Reviewer #4: No

---

## [Editor Report · Decision Letter 3]

21 Dec 2022

Factors driving underweight, wasting, and stunting among urban school aged children: Evidence from Merawi town, Northwest Ethiopia.

PGPH-D-22-00367R3

Dear Mr. Kamal,

We are pleased to inform you that your manuscript 'Factors driving underweight, wasting, and stunting among urban school aged children: Evidence from Merawi town, Northwest Ethiopia.' has been provisionally accepted for publication in PLOS Global Public Health.

Best regards,

Jitendra Kumar Singh, PhD

Academic Editor